# Molecular Mechanisms of Rett Syndrome: Emphasizing the Roles of Monoamine, Immunity, and Mitochondrial Dysfunction

**DOI:** 10.3390/cells13242077

**Published:** 2024-12-17

**Authors:** Julia Lopes Gonçalez, Jenny Shen, Wei Li

**Affiliations:** 1Department of Neurobiology, University of Alabama at Birmingham, Birmingham, AL 35294, USA; jgoncalez@uab.edu (J.L.G.); jennyshen@uabmc.edu (J.S.); 2Graduate Program in Behavioral Neuroscience, University of Alabama at Birmingham, Birmingham, AL 35294, USA

**Keywords:** Rett syndrome, MeCP2, monoamine, immune cell, mitochondria

## Abstract

Rett syndrome (RTT), which predominantly affects females, arises in most cases from mutations in the *Methyl-CpG-binding Protein-2* (*MECP2*) gene. When MeCP2 is impaired, it disrupts the regulation of numerous genes, causing the production of dysfunctional proteins associated with various multi-systemic issues in RTT. In this review, we explore the current insights into molecular signaling related to monoamines, immune response, and mitochondrial function, and their implications for the pathophysiology of RTT. Research has shown that monoamines—such as dopamine, norepinephrine, epinephrine, serotonin, and histamine—exhibit alterations in RTT, contributing to a range of neurological symptoms. Furthermore, the immune system in RTT individuals demonstrates dysfunction through the abnormal activity of microglia, macrophages, lymphocytes, and non-immune cells, leading to the atypical release of inflammatory mediators and disruptions in the NF-κB signaling pathway. Moreover, mitochondria, essential for energy production and calcium storage, also show dysfunction in this condition. The delicate balance of producing and scavenging reactive oxygen species—termed redox balance—is disrupted in RTT. Targeting these molecular pathways presents a promising avenue for developing effective therapies.

## 1. Introduction

Rett syndrome (RTT) is a progressive neurodevelopmental disorder that predominantly affects females, with an incidence of 1:10,000-15,000 births [1,2,3]. Typically, individuals with RTT develop normally until about 6–18 months of age, at which point they begin to manifest a host of neurological and psychiatric symptoms. This condition impacts multiple systems in the body, including the nervous, musculoskeletal, gastrointestinal, respiratory, cardiovascular, urinary, endocrine, and immune systems [4]. In the central nervous system (CNS), RTT disrupts brain development and cognitive ability and causes severe language and communication problems [5]. It also affects the peripheral nervous system, including the autonomic nervous system, resulting in irregularities in breathing, heart rate, gut function, sleep, and body temperature [6,7,8]. Decreased hand mobility, difficulty in walking, and stereotypical hand movements are also present in RTT, which suggest potential musculoskeletal impairments [9]. Gastrointestinal complications are also prevalent, with many patients suffering from issues such as gastro-esophageal reflux disease (GERD), constipation, and abnormal colon dilation [10]. In the respiratory system, RTT patients manifest irregular breathing patterns, including episodes of hyperventilation, breath-holding, the swallowing of air, and apnea [11,12]. Cardiac issues are another concern, as some RTT individuals may experience cardiac arrhythmia and irregular heartbeat, which are risk factors for subsequent sudden cardiac death [8,13]. Additionally, urinary issues such as frequent urinary tract infections, kidney stones, and urine retention are commonly reported among RTT patients [14]. Endocrine system abnormalities impacts growth, bone health, thyroid function, puberty onset, and weight [15]. Finally, a dysfunctional immune system also plays a role in RTT etiology [16].

Most RTT cases are known to stem from mutations in the *methyl-CpG-binding protein 2* (*MECP2*) gene [17]. This gene is essential for regulating the expression of numerous genes in the brain and beyond, resulting in significant disruptions in cellular function that can lead to various multi-systemic comorbidities [18]. Some of the affected molecules are vital for cell differentiation, communication, and neural circuitry, while others are crucial for energy provision and homeostasis maintenance. The importance of the role of growth factors in RTT has been thoroughly examined in the literature [19,20]. Additionally, the molecules involved in glutamatergic and GABAergic pathways have been discussed in the past for their importance in maintaining the balance between excitation and inhibition [21,22,23]. This review will delve into the current understanding of monoamines, cell immunity, and mitochondrial function in RTT, three domains that are recently receiving increased attention due to their interconnected signaling and metabolic pathways that, when dysfunctional, underpin the molecular mechanisms involved in RTT pathophysiology.

### Methods

To identify relevant literature, searches were conducted using the keywords “Rett” combined with “monoamines” or specific monoamine subtypes such as dopamine (DA), norepinephrine (NE), and serotonin (5-HT), among others. Searches also included terms related to microglia, lymphocytes, macrophages, immune signaling, immunity, inflammation, mitochondria, oxidative stress, and reactive oxygen species. Searches were conducted in PubMed, Scopus, and Academic Search Complete, and included studies from January 1980 to November 2024. The search term variations included keyword combinations and synonyms. Studies were included if they were original research, meta-analyses, or systematic reviews. Studies were excluded if they were methodologically inappropriate. Some external reviews not including RTT findings were included for contextualization. Findings were organized by content to provide an ordered summary.

## 2. Role of Monoamines in RTT

Monoamines are key neurotransmitters that have been implicated in various functions throughout both the central and peripheral systems [24]. They encompass catecholamines, including DA, NE, and epinephrine, along with 5-HT and histamine. Catecholamines are synthesized from the amino acid tyrosine, while 5-HT is from tryptophan and histamine from histidine. All monoamines are packed into synaptic vesicles by the vesicular monoamine transporter (VMAT). The biodegradation of monoamines is carried out by monoamine oxidase (MAO), an enzyme found attached to mitochondria, which utilizes O_2_ as an electron acceptor. There are two forms of MAO: MAO-A and MAO-B. DA can be metabolized by both forms, but NE and 5-HT are only broken down by MAO-A [25]. These chemically related monoamines are known to be involved in homeostasis, cognition, movements, emotions, and many other physiological and psychological functions. RTT patients present substantial alterations in the levels of these monoamines and their metabolites (Table 1) [26,27,28].

### 2.1. Dopamine (DA)

DA is a main neurotransmitter in the CNS that influences a variety of behaviors, including motility, food intake, and cognition, among others. These DA functions are primarily mediated through five known types of metabotropic DA receptors: D1–5 [43]. The dopaminergic system is widely distributed throughout in the brain, with a dense neuron population in the substantia nigra (SN) and ventral tegmental area (VTA). These regions primarily send axonal projections to the striatum and nucleus accumbens (NAc) [44]. The dopaminergic system plays a crucial role in several pathological conditions such as Parkinson disease, schizophrenia, and substance use disorders, making it an important target for pharmacological intervention.

The dopaminergic system was an early point of interest in research concerning RTT patients. A study found that levels of the DA metabolite homovanillic acid (HVA) were significantly lower in the cerebrospinal fluid (CSF) of RTT patients compared to healthy controls [29]. However, a similar study could not demonstrate the same difference [45]. Further analysis through postmortem RTT brain studies revealed a decrease in DA levels across various brain regions [31]. Several other studies also indicated that in RTT patients or postmortem brain tissue, the levels of D2 receptors and the DA transporter (DAT) showed alterations in the caudate nucleus and putamen, while D1 receptors appeared unaffected [32,33,34,46,47]. Collectively, these early findings suggest that DA may play a significant role in RTT pathophysiology.

The dopaminergic dysfunction was later tested in *Mecp2*-deficient mice. Synaptic deficits were observed in dopaminergic neurons in *Mecp2*-null mice, with reduced cell capacitance, shorter dendritic lengths, and decreased DA release [35]. Additionally, this mouse strain showed lower levels of HVA and DA when compared to wildtype (WT) animals [30]. In a study by Panayotis and colleagues [36], motor dysfunction was documented in *Mecp2*-null male mice, revealing a decrease in the number of dopaminergic neurons expressing tyrosine hydroxylase (TH) in the SN. Notably, the chronic oral administration of L-Dopa mitigated the motor deficits, highlighting the involvement of DA in the motor issues associated with MeCP2 deficiency. A more recent investigation found that the dopaminergic dysfunction observed in RTT patients is mirrored in *Mecp2*-deficient mice, as both groups showed a reduced density of striatal D2 receptors [37].

Like RTT patients, mouse models often present breathing difficulties [12,48]. Treatment with quinpirole, an orthosteric agonist for D2-like receptors, was effective in reducing both apnea occurrences and respiratory rates in mice [49]. Despite these findings, the cellular and molecular mechanisms that link DA to the symptoms of RTT remain largely unexplored. More research is essential to identify potential treatments that could alleviate the symptoms related to dopaminergic dysfunction.

### 2.2. Norepinephrine (NE)

NE is both a neurotransmitter and a hormone that plays a crucial role in the CNS and peripheral regions through its interaction with adrenergic receptors α and β. The noradrenergic system is involved in a range of complex functions, including arousal, cognition, and attention. Noradrenergic neurons were also found to be essential to the maturation process of the respiratory network in neonatal mice [50,51]. Within the brain, the locus coeruleus (LC) serves as the primary source of NE, extending its extensive projections throughout the brain [52]. Research has demonstrated that activating neurons in the LC through optogenetic methods can trigger NE release, ultimately enhancing global network connectivity [53].

In a similar vein to DA, the NE metabolite 3-methoxy-4-hydroxyphenylethylene glycol (MHPG) has been observed at reduced levels in RTT patients [28,29]. Notably, *Mecp2*-null mice show a 25% decrease in NE levels when compared to their control counterparts [38]. Given that nearly all severe RTT patients experience breathing dysfunction over their lifespan, restoring noradrenergic signaling represent a promising neuropharmacological approach to mitigate respiratory deficits [12]. In this context, treatment using desipramine, a selective NE reuptake inhibitor, has demonstrated improvements in breathing patterns in a mouse RTT model [54,55]. However, it is worth noting that desipramine did not yield significant improvements in the apnea-hypopnea index (AHI) during a phase 2 clinical trial [56]. Our understanding of the NE system in RTT is still quite limited. To improve the quality of life for patients, it is crucial to delve deeper into the mechanisms underlying the NE decline in RTT and to further explore new noradrenergic targets in this research effort.

### 2.3. Epinephrine

Epinephrine, much like NE, is a monoamine that acts as both a neurotransmitter and a hormone. It plays an important role in the autonomic nervous system, specifically in the sympathetic division, where it interacts with the adrenergic receptors α and β. Although epinephrine is less prevalent in the brain compared to other catecholamines, it has neuronal cell bodies located in the lateral tegmental system that primarily projects to the brainstem, hypothalamus, thalamus, and spinal cord [57].

Research on epinephrine has been scarce, probably because of its limited function in the CNS. Notably, a postmortem analysis of an RTT patient revealed decreased levels of epinephrine in the hypothalamus [39]. Given its critical role in the autonomic nervous system, patients with *MECP2* mutations were found to exhibit sympathetic imbalance [58]. Likewise, *Mecp2*-null mice presented a reduced catecholamine in sympathetic ganglion and adrenal medulla while secreting more epinephrine, leading to higher plasma epinephrine levels [59]. Such alterations in epinephrine levels very likely contribute to an imbalance between sympathetic and parasympathetic activity in individuals with RTT.

### 2.4. Serotonin (5-HT)

5-HT is a key neurotransmitter that plays a vital role in various functions, including learning, memory, sleep, appetite, and mood [60]. Due to its relevance for the vast array of brain functions, an imbalance of the serotoninergic system has been implicated in numerous psychiatric and neurological disorders. In the brain, 5-HT is synthesized in the raphe nuclei from tryptophan, then taken back up by the 5-HT transporter (SERT), and ultimately broken down by MAO [61]. The serotonergic effects occur when 5-HT binds to its receptors, which encompass six families of G protein-coupled receptors and one ionotropic channel [62].

In the CSF of RTT patients, 5-HT metabolite 5-hydroxyindole acetic acid (5-HIAA) was observed in reduced levels [27,28], with similar results showing diminished 5-HT levels in plasma [40]. These findings have also been replicated in mouse models, where *Mecp2*-null males and *Mecp2* heterozygous (Het) females exhibited lower 5-HT levels [36,42]. Considering these results, exploring the serotoninergic system will provide valuable perspectives for understanding the mechanisms at play in RTT and potentially lead to effective treatment options.

Following this line of investigation, De Filippis and his team observed a reduction in the density of 5-HT7 receptors in the cortex and hippocampus of RTT male mice that express a non-functional truncated MeCP2 (*Mecp2*^308/y^) [41]. Intriguingly, after administering a repeated systemic treatment with a 5-HT7 receptor agonist, they found improvements in behavioral assessments for anxiety, memory, and motor skills. A subsequent study confirmed these positive effects in a female mouse model as well [63]. A similar treatment with the same agonist was also found to mitigate mitochondrial respiratory chain impairment and oxidative phosphorylation deficiencies, along with improving the energy status in the brains of RTT female Het mice [64].

The 5-HT receptor family is also involved in breathing control [65]. Abdala and colleagues revealed that 8-OH-DPAT, a 5-HT1A agonist, was able to decrease respiratory apneas in *Mecp2*-deficient mice [66]. Similarly, it was shown that F15599, a highly selective post-synaptic 5-HT1A agonist, reduced apneas and improved breathing irregularities [67]. Buspirone is a 5-HT1A agonist used in the treatment of anxiety and mood disorders, and has also been investigated in RTT [68]. Using a non-invasive wearable sensor, it was shown that buspirone treatment normalized electrodermal activity and heart rate variability in RTT patients [69]. Tandospirone, another 5-HT1A agonist, is used to treat anxiety and has demonstrated great efficacy in improving social anxiety, motor dysfunction in Parkinson’s disease, and cognitive deficits in schizophrenia [70]. In *Mecp2*-null mice, tandospirone treatment significantly extended lifespan and improved general condition, motor function, and breathing [71]. Mechanistically, tandospirone treatment significantly ameliorated the impairment in GABAergic, glutaminergic, dopaminergic and serotoninergic neurotransmission in the brainstem [71]. Using RNA sequencing analysis, the authors found that the modulation in RTT phenotype is partially through rescuing the CREB1/BDNF signaling pathway [71]. Taken together, these findings suggest that activating 5-HT1A could be beneficial for the treatment of RTT, particularly for patients experiencing respiratory impairments.

Since 5-HT is involved in the regulation of motor activity, selective 5-HT reuptake inhibitors (SSRI) are a therapy of interest for motor control deficit. Fluoxetine is one such therapy for amelioration of the motor abnormalities often seen in RTT patients. Villani and colleagues studied the effects of fluoxetine on alleviating motor coordination impairments in both male *Mecp2*-null and female *Mecp2* Het mice [72]. Their findings revealed that repeated fluoxetine administration in female mice completely reversed rotarod performance deficits, which could be abolished by inhibition of 5-HT synthesis. Notably, this inhibition did not affect the concentration of the 5-HT precursor, 5-hydroxytryptophan (5-HTP), in any of the brain regions examined. In contrast, *Mecp2*-null male mice showed no improvement in their rotarod performance, although a decrease in 5-HTP levels was observed in the prefrontal cortex, hippocampus, and striatum [72]. 

These differences in sex are thought to stem from the release of X-inactivating specific transcript (Xist) from the suppression of X chromosome inactivation (XCl) that leads to a mosaic expression of MeCP2 in female Het mice. Fluoxetine has the potential to enhance the expression of TGF-β1, a factor known to inactivate Xist, thereby increasing MeCP2 levels in *Mecp2*-deficient cells. Additional evidence appears to support this possibility. Villani and colleagues investigated MeCP2 expression in the brain of female *Mecp2* Het mice following fluoxetine treatment [72]. Their findings showed an increase in the number of MeCP2-expressing cells in the prefrontal cortex, motor cortex, and lateral striatum, but there was no change observed in the CA3 region of the hippocampus. Despite these positive findings concerning fluoxetine, a more recent study from the same team indicated that simply raising 5-HT through its precursors was not sufficient to address motor coordination deficits, indicating that fluoxetine may rescue motor control deficits through pathways other than mere serotonin level increase [73]. However, while this research shows promising insight into how fluoxetine’s role in modulating 5-HT can aid in ameliorating RTT motor control deficits in human patients, fluoxetine is only sometimes suggested to treat mood disorders in RTT patients, possibly due to complex contraindications with the pathology of RTT [74].

Another antidepressant that could be a candidate as a repurposed drug is mirtazapine. With a well-established safety profile [75], and an absence of anticholinergic and cardiorespiratory side effects [76,77], this noradrenergic and specific-serotonergic tetracyclic antidepressant has been investigated in animal models. Treatment with mirtazapine in *Mecp2*-null mice completely restored dendritic arborization and spine density in somatosensorial pyramidal neurons [78]. Moreover, mirtazapine stabilized blood pressure, breathing pattern, and anxiety levels. Further research showed that chronic treatment in *Mecp2*-deficient female mice was able to maintain motor learning and restore parvalbumin levels in the primary motor cortex and barrel cortex [79]. The same study also investigated RTT adult female patients who were prescribed mirtazapine for insomnia and mood disorders, were evaluated with the motor behavior assessment scale (MBAS) and Rett clinical severity scale (RCSS). Continued treatment slowed the disease progression and led to significant improvements in both MBAS and RCSS [80]. These findings underscore the necessity of understanding the specific mechanisms behind the serotoninergic role in RTT to enable more effective therapeutic interventions.

The serotoninergic system also influences blood pressure, although the underlying mechanisms remain unclear [81]. Blood analysis of untreated RTT patients displayed lower 5-HT concentrations compared to patients treated with anticonvulsant drug [40]. In untreated patients, 5-HT concentrations were correlated with sympathovagal balance, a relationship not observed in treated patients. A radioligand approach for SERT in the dorsal motor nucleus of the vagus showed that SERT binding decreased over time in control postmortem tissue, with no such change in RTT [82]. When adjusted for age, binding differences between the controls and RTT were found to be statistically significant.

### 2.5. Histamine

Histamine functions as both a neurotransmitter and a hormone, engaging with four G protein-coupled receptors (H1–H4), with the first three expressed in the CNS [83]. Histaminergic neurons are located in the hypothalamus with projections throughout the brain [84]. While histamine has a range of functions in the periphery and the brain, this neurotransmitter acts mostly as promoting wakefulness and suppressing rapid eye movement (REM) sleep [85].

The potential involvement of histamine in RTT is still unclear, as this connection has not been directly explored. Nonetheless, RTT patients often experience multiple sleep disorders, a strong indication of histamine involvement [86].

### 2.6. Implications on the Immune System

Monoamines are known to function as a signaling molecule between the nervous system and immune system, and between individual cells of the immune system [87]. The presence of DA, for example, inhibits production of angiotensinogen in astrocytes and inhibits the angiotensin II/NADPH oxidase pro-inflammatory axis in microglia [88]. Amongst natural killer (NK) cells, increased serotonin levels decrease inhibition of NK cell cytotoxicity and cytokine production; conversely, the presence of epinephrine and norepinephrine, which bind to α- and β-adrenergic receptors on NK cells, causes decreased cell cytotoxicity and cytokine production [89]. Therefore, monoamine imbalances in RTT could be implicated in immune system dysfunction, in addition to the molecular signaling issues caused by MeCP2 deficiency in immune cells.

## 3. Immune and Inflammatory Mechanisms in RTT

Growing evidence demonstrates that dysfunctional immunity and subclinical inflammation are the significant pathophysiological mechanisms contributing to the progression of RTT (Figure 1) [16,80,90]. These mechanisms engage a range of immune cells, including microglia, macrophages, and lymphocytes, along with non-immune cells like astrocytes. Research has uncovered changes in molecular immune signaling in individuals with RTT and in mouse models of this disorder.

### 3.1. Microglia

Microglia, the innate immune cells of the brain, are the primary parenchyma-resident macrophages originating from yolk sac precursors [91]. Their role in RTT development and progression has garnered significant attention [92,93]. A pioneering study revealed that restoring WT microglia through bone marrow transplantation or reintroducing *Mecp2* using Cre-Lox recombination in various myeloid cells led to a prolonged lifespan and reduced RTT-like symptoms in male *Mecp2*-null or female Het mice [94]. Later research demonstrated that re-expressing MeCP2 specifically in microglia also extended the lifespan [95]. However, these encouraging results were not able to be replicated in a subsequent study [96]. Additionally, research has questioned the role of microglia during development, with findings indicating that microglia-mediated synaptic pruning in the retinogeniculate system occurs independent of MeCP2 [97]. Such conflicting results have propelled numerous investigations into the molecular, cellular, morphological, and behavioral aspects of microglia in the context of RTT.

General morphological analysis of *Mecp2*-null mice showed that they had fewer numbers of microglia, a soma that was initially smaller but later increased in size, and a reduced process branching [95] (Figure 1A_1_). Consistent with the altered morphology, a smaller soma size was also observed in microglia-like cells derived from RTT patient iPSCs [98]. Furthermore, a lack of MeCP2 in microglia resulted in an altered expression of genes encoding for molecules involved in cell movement, adhesion, inflammation, and phagocytosis [95,99,100]. Indeed, the microglia-like cells or microglia from *Mecp2*-null mice exhibited lower viability, impaired migration, and decreased phagocytic ability [94,99,100]. *Mecp2*-lacking microglia stimulated with the pro-inflammatory factors display a distinct gene expression, indicating their role in regulating inflammatory gene transcription [95,100,101].

Furthermore, *Mecp2* deletion affects synapses and dendrites of cultured hippocampal neurons and iPSCs-derived microglia, which may be mechanistically caused by excitotoxicity during excessive glutamate release (Figure 1A_2_) [100,102]. The increase in glutamate release can be linked to the overproduction of glutamate due to the heightened expression of the microglia-specific glutamine transporter, known as sodium-coupled neutral amino acid transporter 1 (SNAT1). Increased SNAT1 activity within the glutamate synthesis pathway occurs due to a lack of MeCP2-mediated repression [103]. The receptor-interacting protein kinase 1 (RIPK1) in microglia could also be involved in excessive glutamate release. It plays a role by triggering oxidative stress and cytokine release, which can impact the transporters responsible for glutamate synthesis [104]. Cysteine–glutamate antiporters that function to export glutamate to the extracellular space while importing cysteine may also contribute to the higher concentration of extracellular glutamate, as they are upregulated in *Mecp2*-null microglia [101].

### 3.2. Peripheral Macrophages

Peripheral macrophages, which originate from monocytes, circulate in the blood and can also be located at the boundary of the brain and the perivascular spaces (Figure 1B_1_) [105]. An early study revealed that mice lacking MeCP2 in macrophages developed obesity, although they did not show any noticeable neurological symptoms [106]. In a *Mecp2*^308/y^ RTT animal model, there was a notable increase in the number of macrophages located at the interface of the periphery and parenchyma. Furthermore, it has been observed that TNF-α expression is elevated in peripheral blood mononuclear cells (PBMCs) [107]. Conversely, MeCP2 deficiency has been linked to a decrease in TNF-α expression and a reduced sensitivity to inflammatory stimuli in macrophages (Figure 1B_2_) [108]. Further investigations are needed to determine if TNF-α expression is region-specific.

The dendritic spine abnormalities and social deficits seen in *Mecp2*^308/y^ mice might be related to the overactivation of purinergic P2X7 receptors that are known to be involved in neuroinflammation (Figure 1B_2_) [109]. When these receptors were pharmacologically blocked or genetically removed, there was a reduction in macrophage density as well as improvements in both morphological and cognitive dysfunction [110]. The molecular processes behind these defects linked to P2X7 receptors may involve an excessive outflow of K^+^ and an influx of Ca^2+^ and Na^+^, leading to the release of inflammatory cytokines like IL-1β, IL-6, and TNF-α [111].

### 3.3. Lymphocytes

The primary peripheral lymphocytes include T cells, B cells, and NK cells [112]. T cells can be categorized into CD8+ cytotoxic and suppressor cells, as well as CD4+ helper cells. An early investigation involving a limited number of RTT cases found no significant change in T and NK cell numbers in blood samples [113]. However, later research with a larger sample indicated a reduction in CD8+ T cells and NK cells [114]. In contrast, these two cell types were observed to be increased in the meninges of *Mecp2*-null mice (Figure 1B_1_). These findings suggest that the immune status may vary in different regions in RTT [115].

Gene expression profiling in clonal lymphocyte cultures derived from patient T cells showed no substantial changes; however, these cells demonstrated a diminished response to immune stimulation [116,117]. In *Mecp2*-deficient CD4+ T cells from conditional KO mice, it was noted that expression of the gene miR-124—crucial for inhibiting the translation of suppressor-of-cytokine signaling 5 (Socs5)—was decreased (Figure 1B_3_). This reduction contributed to the lower cytokine production (such as IFN-γ) and hindered the differentiation into T helper cells [118]. Moreover, MeCP2 is essential for the differentiation of CD4+ T cells into suppressor cells through the regulation of the expression of the Forhead box P3 (Foxp3) transcription factor [119]. Contrasting the diminished IFN-γ in conditional *Mecp2* KO CD4+ T cells, an increase was noted in the meninges of *Mecp2*-null mice [115]. Several studies have also reported an increase in the release of various pro-inflammatory mediators. The activation of CD4+ T cells resulting from conditional *Mecp2* KO coincides with heightened IL-17 production [119]. PBMCs obtained from RTT patients displayed increased secretion of cytokines (including IL-8, IL-9, and IL-13) and evaluated levels of the oxidized fatty acid 13-HODE [120,121], whereas soluble IL-2 receptors were also founded to be augmented in patient serum [114].

The regulation of genes within B cells by MeCP2 has also been established [122]. Research in B cell populations has shown either unchanged or increased cell density across various samples [113,114,115]. Furthermore, there is evidence indicating that RTT may involve an autoimmune component related to B cell function [123]. In blood samples from RTT patients, the serum levels of autoantibodies targeting nerve growth factors were found to be significantly increased (Figure 1B_1_) [124]. Similarly, elevated autoantibodies against folate were observed in some populations with RTT, potentially leading to cerebral folate deficiency [125]. Additionally, the presence of serum IgM autoantibodies against *N*-glycosylated components could interfere with proper protein glycosylation [126].

### 3.4. Inflammatory Mediators

Cytokines and chemokines are key mediators in immune responses in RTT, significantly influencing neuronal function [16]. When comparing RTT patients to healthy individuals, measurement of these mediators revealed varying levels: some cytokines, like TNF-α, IL-4, IL-5, IL-6, IL-8, IL-17A, and IL-33, show increased concentration, while others, such as IFN-γ, IL-12p70, TGF-β1, IP-10, I-TAC, and RANTES, exhibit decreased levels. However, several mediators, including IL-1β, IL-10, and IL-13, remain unchanged [127]. Notably, a separate study found significant increases in only IL-8, IL-9, and IL-13 within RTT serum, while mediators like IFN-*γ* did not show any increase [120]. The varying secretion levels of these mediators can be attributed to numerous factors, including the source of the samples, the activity status of immune cells, and stages of disease progression. Furthermore, saliva samples from RTT patients reveal heightened concentrations of IL-β, IL-6, IL-8, IL-10, and TNF-α, which correlate with the severity of RTT [128].

Cytokines and chemokines are produced and released by immune and non-immune cells. Research shows that lipopolysaccharide (LPS) stimulation in primary *Mecp2*-null microglia led to an increased release of inflammatory factors like TNF-α, IL-10, INF-γ, and IL-12 (Figure 1B_4_) [101]. However, in iPSC-derived microglia lacking MeCP2, the release of these factors was not significantly altered, except for MIP-1α [100]. Interestingly, inhibiting the CX3CL receptor CX3R1 in microglia was found to reverse neuronal anomalies and alleviate RTT-like symptoms [129]. In *Mecp2*-null zebrafish, the levels of IL-1β and IL-10 mRNA were increased, whereas TNF-α was downregulated [108]. In human *Mecp2*-lacking PBMCs, the levels of TNF-α, IL-3, IL-6, IL-8, IL-13, and IL-15 expression levels were found to be enhanced, with the exception of IL-9 [107,120]. Notably, these inflammatory mediators can also arise from mucosal immune cells affected by gut microbiota [130].

The microbiota–gut–brain axis (MGBA) acts as a communication network between the brain and the gut and has implications in immune system dysfunction in RTT patients. The gut microbiota in RTT patients is less diverse, with altered numbers, which is partially responsible for clinical symptoms such as constipation and long-term low-grade gut inflammation [131]. Gut bacteria are involved in neurotransmitter metabolism and modulation through the metabolism of short-chain fatty acids (SCFAs) [132]. A small number of studies have found altered SCFA metabolic pathways in the gut bacteria of human RTT patients, which may affect neurotransmitter availability [12]. The gut is also directly affected by CNS modulation of the intestinal tract permeability and mobility [131]. *Mecp2*-null mice exhibit impaired colon epithelium [133], and this dysfunctional intestinal morphology, paired with the altered changes in microbial abundance and diversity, may contribute to cytokine dysregulation in RTT [132,134]. Moreover, RTT patients have shown a persistent T cell population and elevated levels of IL-1β and IL-10 due to gut fungi infection [135]. Interestingly, a study on male and female RTT mice demonstrated a between-sex distinct cytokine profile in their fecal samples [136]. While both sexes showed reduced levels of IFN-γ, male mice consistently had lower IL-4 concentrations throughout the disease progression, unlike female mice, which displayed normal levels initially and increased levels at later stages. This difference in cytokine profiles hints at a unique role for pro-inflammatory factors in the pathophysiology of RTT across the sexes. Beyond immune cells, non-immune cells are also capable of releasing cytokines and chemokines [137]. For instance, cultured *Mecp2*-null astrocytes influenced the synaptogenesis of co-cultured WT cells by secreting IL-6, and this effect could be mitigated by specific scavenger antibodies [138].

These findings provide compelling evidence that targeting inflammatory mediators could be a promising strategy for developing effective treatments. Notably, a recent study showed that transplanting neural precursor cells (NPCs) into the brains of *Mecp2*-deficient mice led to improvements of RTT-like symptoms [139]. This discovery points to the activation of the IFN-γ pathway as a crucial underlying mechanism, which is further supported by the demonstrated effectiveness of directly injecting IFN-γ into the cerebrospinal fluid. RTT patients often present with impaired bowel movement and gastrointestinal distress caused, in part, by chronic inflammation. Imaging has also shown that RTT patient lungs show signs of alveolar inflammation and thickened bronchiolar walls, which may indicate that inflammation has a role in the abnormal breathing phenotypes found in RTT. Selectively modulating the implicated inflammatory mediators could help manage such symptoms [80].

### 3.5. NF-κB Signaling Pathway

Ample evidence highlights the significant role of the transcription factor NF-κB in RTT. Studies have shown that the deletion of MeCP2 from PBMCs, the human monocyte line THP1, or the mouse cortex led to an increase in NF-κB expression (Figure 1B_5_) [107,140,141]. NF-κB activation in these contexts resulted in elevated expression of inflammatory molecules such as TNF-α, IL-3, and IL-6, along with increased glutamate release. In RTT fibroblasts, the nuclear translocation of NF-κB and the expression of its downstream cytokine IL-1β remain persistently high and do not respond appropriately to the pro-inflammatory stimulation [142]. This dysregulation is associated with a constitutive activation of the NOD-, LRR- and pyrin domain-containing protein 3 (NLRP3) inflammasome and other inflammatory regulators [143]. In studies involving *Mecp2*-null mice, inhibiting NF-κB signaling has been shown to restore dendritic structure and extend the lifespan [140]. Furthermore, NF-κB signaling operates downstream of the glycogen synthase kinase-3β (GSK-3β) [144]. An inhibition of GSK-3β reduced NF-κB activity, leading to the reduced secretion of inflammatory mediators such as IL-1, IL-4, IL-12p70, and IL-17. This reduction in NF-κB signaling mirrors the beneficial effects seen with the decrease in GSK-3β activity, correcting dendritic abnormality, improving motor defects, and extending the lifespan. Moreover, dietary supplementation with vitamin D, known to mitigate aberrant NF-κB signaling, has been effective in restoring impaired dendritic structure and lengthening the lifespan of *Mecp2*-null mice [145]. Vitamin D treatment also restores deregulated genes in *Mecp2* HT mice, rescuing the abnormal neuronal morphology and enhancing motor and cognitive abilities [146]. Collectively, these findings lay a molecular foundation for further exploration of NF-κB signaling in context of RTT immunity.

## 4. Mitochondrial Dysfunction in RTT

Mitochondria are organelles primarily involved in ATP production, but also play essential roles in other critical processes like calcium storage, apoptosis, and notably, immune response and inflammation [147]. For example, mitochondria themselves can function as immune system ligands due to their separate bacterial ancestry; and notably, breakdown of cells due to mitochondrial dysfunction can lead to higher levels of extracellular ATP, which then may bind to purinergic receptors such as P2X7 and P2Y2, leading to inflammatory responses [148]. The human mitochondria, while having its own genome, also depends on the nuclear genome, several of which are regulated by MeCP2 [149,150]. Consequently, mutations in the *Mecp2* gene can lead to direct impairments in the aforementioned mitochondrial functions [151]. Studies investigating mitochondrial dysfunction in RTT have been conducted for many years (Figure 2) [152,153,154].

### 4.1. Redox Balance

Mitochondria play a crucial role in cellular metabolism as the main site for the production and scavenging of reactive oxygen species (ROS) [155]. During the process of ATP production, electrons might leak from the electron transport chain (ETC) and react with oxygen to form superoxide (O_2_^−^). While ROS are essential to cell signaling [156], elevated levels can cause cellular damage. Hence, the process of scavenging, which involves the removal of excess ROS, is crucial in maintaining normal cellular function. Once O_2_^−^ is formed, it can either rapidly react with nitric oxide or be reduced to hydrogen peroxide (H_2_O_2_) through the action of superoxide dismutase (SOD). H_2_O_2_ is then converted into water and oxygen by catalase or glutathione peroxidase (GPx). However, when ROS production exceeds normal levels and detoxification processes falter, a condition known as redox imbalance occurs, leading to oxidative stress (OS) [157]. RTT has been associated with mitochondrial dysfunction due to the occurrence of redox imbalance in the brains of presymptomatic *Mecp2*-deficient mice [158]. Several studies have indicated an increase in OS markers in RTT patients and mouse models, coupled with a diminished scavenging system.

### 4.2. Impaired Mitochondria in RTT

Early signs of mitochondrial dysfunction in RTT emerged from a comparison of postmortem brain tissue between a single patient and a healthy control. In this study, Sofić and colleagues discovered reduced concentrations of antioxidants, specifically ascorbic acid (AA) and glutathione (GSH) [159]. AA plays a role in neutralizing ROS oxidized by donating electrons, whereas GSH serves as a hydrogen donor for GPx to reduce H_2_O_2_. Subsequent research has corroborated these initial data, revealing consistently lower levels of antioxidant molecules in the brain and serum of RTT patients [160,161]. The activity of superoxide dismutase 1 (SOD1), an enzyme primarily located in the cytoplasm and the intermembrane space of mitochondria, plays a crucial role in maintain redox homeostasis [162]. However, research has shown that its activity was reduced in patients with RTT compared to control groups [163].

Recent investigations into the mitochondrial ETC in RTT gained momentum with the introduction of animal models. The ETC consists of four main protein complexes (I–IV), with H_2_O_2_ production primarily arising from complexes I and III in WT mice [164]. However, female *Mecp2* Het mice exhibit a dysfunction in complex II, resulting in an increase in H_2_O_2_ production [165]. Interestingly, treatment with bacterial cytotoxic necrotizing factor 1 (CNF1) that has previously shown positive effects in behavior and bioenergetic markers [166], resulted in the reactivation of the respiratory chain, effectively preventing the rise in H_2_O_2_ concentrations. A study employing the quantitative redox sensor roGFP1 also detected more oxidized conditions in the cytosol of hippocampal neurons from *Mecp2*-deficient mice compared to WT controls [167]. In human studies, data from twenty-seven RTT patients revealed an upregulation of metallothionein in blood, which correlated directly with the severity of the symptoms [168]. Another indication of mitochondrial dysfunction is the number of copies of mitochondrial DNA (mtDNA) [169]. Real-time quantitative polymerase chain reaction (RT-PCR) analyses demonstrated an increase in mtDNA copies in RTT patients compared to healthy controls [170].

More recently, advancements in human embryonic stem cell (hESC) lines, along with bioinformatics and proteomics, have enabled more specific investigations to elucidate OS in RTT. Transcriptome analysis have revealed that *Mecp2* mutations affect gene expression in astrocytes, impacting their ability to interact with neurons [171]. In a study using hESC derived astrocytes lacking MeCP2 function, Tomasello and colleagues observed reduced mitochondrial respiration and altered protein in the ETC when compared to control astrocytes [172]. Furthermore, these dysfunctional mitochondria exhibited an increase in ROS, and when these mitochondria were transferred to neurons, they influenced normal neuronal activity. This suggests a potential connection between astrocytic mitochondrial impairment and the pathophysiology of RTT [172].

Cicaloni and colleagues performed a protein profiling study using dermal fibroblasts from controls and RTT patients [173]. Their proteomic analysis uncovered significant changes in the expression of proteins associated with mitochondrial structure and function, as well as those involved in cellular stress response. Notably, those proteins related to mitochondrial organization and cell growth were found to be downregulated in RTT patients, whereas proteins linked to antioxidant functions were upregulated, potentially suggesting a compensatory response. It has been known that different mutations in *Mecp2* gene are related to different phenotypes and severity levels of RTT [174]. An independent study using dermal fibroblasts was able to relate different *Mecp2* mutations to mitochondrial proteomic changes [175]. By taking advantage of a label-free quantification analysis through mass spectrometry, they compared health individuals with patients harboring two distinct *Mecp2* mutations. The study confirmed a redox imbalance in RTT samples, revealing that SOD1 was present exclusively in the controls. SOD1 or many other proteins are known to be regulated by nuclear factor erythroid 2-related factor 2 (NRF2) [176]. Examination of those proteins regulated by NRF2 indicated that each RTT subgroup exhibited varying protein expressions. This variation in protein regulation among RTT fibroblasts with different *MECP2* gene mutations implies the presence of distinct mechanisms for oxidative stress defense [175].

Differentially expressed proteins were also observed in *Mecp2*-null mice. Using two-dimensional gel electrophoresis and mass-spectrometry, a study identified a range of differentially expressed mitochondrial proteins in the hippocampus and neocortex [177]. Proteins involved in electron transport and ATP synthesis were upregulated in the hippocampus alone or in both regions. Conversely, proteins essential for mitochondrial dynamics and stress defense showed reduced levels in the neocortex. These protein expression changes shed light on altered mitochondrial function and structure in RTT, emphasizing the distinct mitochondrial proteome changes in different brain regions and suggesting a possible metabolic element to the disorder. *Mecp2*-null male mice exhibit more severe conditions in adulthood than their female Het counterparts. Using this animal model, researchers identified one hundred and one deregulated metabolites in the cortex, especially those related to carbohydrate and amino acid metabolism, when compared to WT controls [178]. Similarly, a study involving humans examined the plasma metabolite profiles of RTT patients alongside their healthy, same age and sex, and siblings [179]. This investigation showed sixty-six altered metabolites, with almost half of them linked to amino acids, suggesting that these differences might reflect underlying pathological processes associated with RTT.

### 4.3. Mitochondrion-Targeted Therapies in RTT

Given the severity of mitochondrial dysfunction and its clear association with RTT, identifying potential therapeutic targets is crucial. A study using a wide range technique found that fibroblast samples from RTT patients displayed significant changes in mitochondrial shape and bioenergetics. The analysis revealed simplified and less interconnected mitochondrial networks, decreased ATP production, and increased oxidative stress [180]. Similar alterations are observed in female mouse models. Building on this, the authors explored the effects of leriglitazone, a selective peroxisome proliferator-activated receptor gamma (PPARγ) agonist, to see if the long-term administration over seven months could ameliorate these bioenergetic deficits. PPARγ is a nuclear receptor that regulates expression of genes associated with mitochondrial biogenesis and antioxidant responses [181]. In all evaluated RTT models, leriglitazone not only corrected mitochondrial dysfunction, but also enhanced the phenotypic outcomes in symptomatic RTT female mice and mitigated the neuroinflammatory components linked to RTT pathophysiology [180]. These promising results lay a strong groundwork for further investigation into the use of leriglitazone and similar treatments in clinical trials aimed at addressing the metabolic issues related to RTT.

The Sigma-1 receptor is a molecule of interest for specifically targeting redox imbalance, as it is protective against OS and activates antioxidant systems [182]. It was shown that the activation of Sigma-1 receptor improved OS in cultured astrocytes, due to the downregulation of iNOS and TNF-α expression, as well as the upregulation of GSH [183]. Blarcamesine is both a Sigma-1 receptor agonist and a muscarinic receptor modulator Treatment with blarcamesine ameliorated sensory and motor phenotypes in female *Mecp2* Het mice [184]. A phase 3 clinical trial (NCT03941444) revealed positive results with no major adverse effects [185]. However, a separate clinical trial (NCT04304482) showed no notable differences between blarcamesine and placebo [185].

Metformin is a widely used medication for the treatment of type 2 diabetes. Although its exact mechanisms of action remain unclear, it is believed to target mitochondrial dysfunction by enhancing scavenging systems [186]. A study investigated whether metformin treatment could reverse brain mitochondrial dysfunctions and OS in female *Mecp2* Het mice [187]. The treatment resulted in increased ATP levels in the whole brain, reduced OS, and improved blood ROS levels. However, there was no improvement in overall health or motor impairments. A subsequent study by the same group showed similar results with chronic treatment [188]. More recently, chronic treatment was found to induce MeCP2 expression in the hippocampus in a sex-specific manner [189]. Considering that metformin is a well-established medication, even for pediatric patients, these results are significant for potential drug repositioning for RTT.

Another study revealed that cannabinoid receptor 1 (CB1R) is overexpressed on the brain mitochondrial membrane of female *Mecp2* Het mice [190]. Notably, systemic administration of rimonabant, a CB1R inverse agonist, successfully normalized this overexpression and restored mitochondrial dysfunction. Given the lower levels of antioxidants, we might consider targeting an increase in these levels to restore the scavenging system. To explore this idea, Baroncelli and colleagues conducted a preclinical trial involving both male and female *Mecp2*-deficient mice, administering vitamin E, *N*-acetylcysteine, and α-lipoic acid orally [191]. While this treatment did not lead to noticeable changes in behavioral performance, breathing patterns, or life expectancy, it did improve hippocampal synaptic plasticity, reduce glucose levels, and ameliorate microcephaly features. Additionally, in healthy individuals, the well-known antioxidant coenzyme Q10 (CoQ10) has been shown to boost ATP production and exhibit effective antioxidant properties, which may help ameliorate mitochondrial deficits [192]. In the pilot study, RTT patients were given a daily CoQ10 supplement over a period of 12 months. The results showed a reduction in OS-induced damage, indicating that CoQ10 treatment could potentially reduce OS damage in erythrocytes.

## 5. Concluding Remarks

The impact of RTT on nearly all systems of the human body is linked to MeCP2 protein dysfunction. When impaired, this transcriptional regulator disrupts widespread gene expression, influencing critical signaling molecules including monoamines, immune mediators, and mitochondrial elements that contribute to many neurological symptoms in RTT.

While there is ample history of animal model research conducted on the individual roles of monoamines, immune mediators, and mitochondria in RTT, a comprehensive understanding of the molecular mechanisms amongst the three systems has yet to be fully realized. Recent surges in both animal and human model studies focused on the immune system and mitochondria provide promising insights into their roles in neurological disorders, but these relationships have yet to be studied under the context of RTT. Furthermore, the overall findings have yet to create a cohesive understanding due to variable data, fragmented research, and lack of human studies in these areas. As efforts continue to unravel the complex roles of these critical signaling molecules in RTT, there is optimism that the insights gleaned from this research will lead to effective multi-system therapies targeting the basal molecular pathways for those affected by this condition in the near future.

## Figures and Tables

**Figure 1 cells-13-02077-f001:**
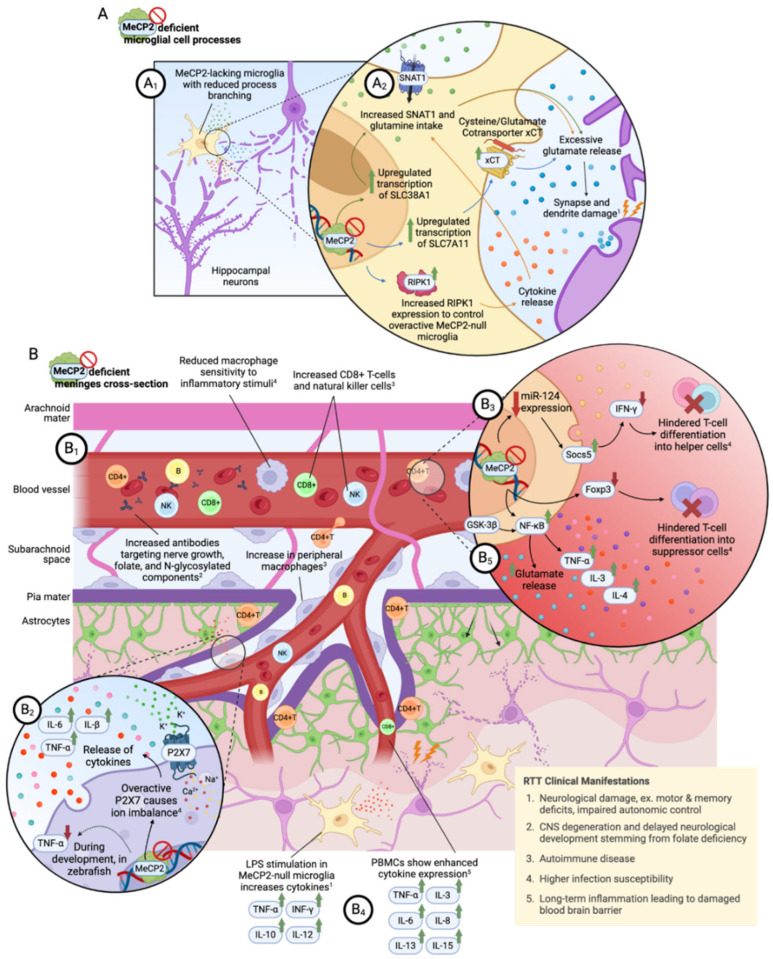
Role of the immune system in RTT. (**A_1_**) MeCP2-lacking microglia show structural and functional deficits compared to WT microglia. (**A_2_**) An overexpression of various gene products leads to enhanced cytokine and glutamate release, causing damage to nearby neurons. (**B_1_**) Simulated cross-section shows various effects of MeCP2 deficiency on immune system dysfunction in the brain meninges. (**B_2_**) Peripheral macrophages with MeCP2 deficiency have an overactive P2X7 receptors, causing ion imbalances that lead to increased expression of cytokines IL-6, IL-β, and TNF-α. MeCP2 deficiency also causes a decreased expression of TNF-α in macrophages during development. (**B_3_**) Mouse CD4+ T cells show decreased miR-124 expression with MeCP2 deficiency. This causes an increased production of suppressor of cytokine signaling 5 (Socs5), thus decreasing overall cytokine production and hindering T cell differentiation into T-helper cells. MeCP2 deficiency also downregulates the Forhead box P3 gene (Foxp3), which is necessary for T cell differentiation into suppressor cells. (**B_4_**) Lipopolysaccharide stimulation in Mecp2-null mouse microglia increases specific cytokine expression. Human peripheral blood mononuclear cells (PBMCs) with MeCP2 deficiency also show increased cytokine expression. (**B_5_**) MeCP2-deficient cells show increased NF-κB activity, leading to an elevated expression of cytokines and glutamate. NF-κB signaling operates downstream of glycogen synthase kinase-β (GSK-3β), providing a target molecule to decrease NF-κB overactivity. The cartoons were created with BioRender.com.

**Figure 2 cells-13-02077-f002:**
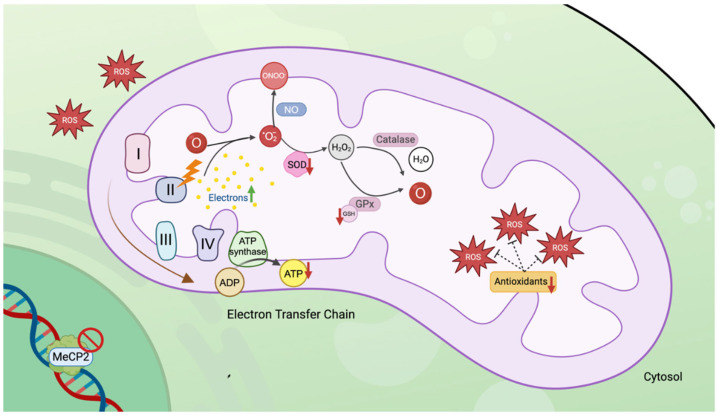
Mitochondrial dysfunction in RTT. The dysfunction of Complex II in the electron transfer chain may increase electron leakage, potentially leading to elevated levels of H_2_O_2_. Superoxide dismutase (SOD) is reduced or absent in patients with RTT. Additionally, lower levels of glutathione (GSH)—which acts as a hydrogen donor to convert H_2_O_2_ into water and oxygen—are associated with diminished antioxidant levels, contributing to heightened production of reactive oxygen species (ROS). The accumulation of ROS in the mitochondria can lead to oxidative stress (OS) within the cell. The cartoon was created with BioRender.com.

**Table 1 cells-13-02077-t001:** Monoamine alterations in different brain regions of RTT patients and *Mecp2*-deficient mice.

Neurotransmitter	Subject	Region	Findings	Reference
Dopamine	Human (RTT)	CSF	↓ HVA	[29,30]
		SFG, STG, OC, PUT	↓ DA	[31]
		Str	↑ D2	[32]
		Str	↓ DAT	[33]
		PUT, Str	↓ D2	[34]
	*Mecp2*-deficient	DA neurons in SN	↓ Cell capacitance ↓ Dendritic length ↓ DA release	[35]
		SN	↓ DA neurons	[36]
		Str	↓ D2	[37]
	*Mecp2*-null	Whole brain	↓ DA	[30]
Norepinephrine	Human (RTT)	CSF	↓ MHPG	[28,29]
	*Mecp2*-null	Whole brain	↓ NE	[30,38]
Epinephrine	Human (RTT)	Hyp	↓ Epinephrine	[39]
Serotonin	Human (RTT)	CSF	↓ HIAA	[27,29,30]
		Plasma	↓ 5-HT	[40]
	*Mecp2*-deficient	Hyp	↓ 5-HT	[36]
		MC, SMC, Pir and Hipp	↓ 5-HTR7	[41]
	*Mecp2*-null	PFC, MC	↓ 5-HT	[42]

Abbreviations: 5-HT, serotonin; 5-HTR7, serotonin receptor 7; CSF, cerebrospinal fluid; D2, dopamine receptor 2; DA, dopamine; HIAA, 5-hydroxyindole acetic acid; DAT, dopamine transporter; Hipp, hippocampus, HVA, homovanillic acid; Hyp, hypothalamus; MC, motor cortex; MHPG, 3-methoxy-4-hydroxyphenylglycol; NE, norepinephrine; OC, occipital cortex; PFC, prefrontal cortex, Pir, piriform cortex; PUT, putamen; SFG, superior frontal gyri, SMC, sensorimotor cortex; SN, substantia nigra; STG, superior temporal gyri; Str, Striatum. ↓, decrease; ↑, increase. For details, refer to the text.

## Data Availability

No new data were created or analyzed in this study.

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
