# Peer review of "Molecular Mechanisms of Rett Syndrome: Emphasizing the Roles of Monoamine, Immunity, and Mitochondrial Dysfunction"

_cells, 2024, doi:10.3390/cells13242077_

Round 1
Reviewer 1 Report
Comments and Suggestions for Authors
The Manuscript by Goncalez et al. explored how the signaling molecules such as monoamines, immune mediators, and mitochondrial elements are involved in the clinical manifestations of Rett syndrome (RTT), a progressive neurodevelopmental disorder caused by mutations in the Methyl-CpG-binding Protein-2 (MECP2) gene. The manuscript in the present form is not suitable for the publication. I encourage the AA to improve all the manuscript sections and, for this reason, I would recommend a major revision.
Major comments
1) Lines 537-538. The AA describe the single contribution of these signaling molecules to RTT but did not explain whether a common causal link exists between monoamines, immunity/inflammation, and mithocondrial dysfunction. The AA should make some efforts in digging in the understanding of the molecular mechanisms at play.
2) Lines 353-364. “Notably, these inflammatory mediators can also arise from mucosal immune cell affected by gut microbiota……”. The MGBA represents a sophisticated communication network between the brain and the gut, involving immunological, endocrinological, and neural mediators. Evidence shows that alterations in gut microbiota composition, or dysbiosis, significantly impact neurological disorders like anxiety, depression, autism, Parkinson's disease (PD), and Alzheimer's disease (AD). Dysbiosis can affect the central nervous system (CNS) via neuroinflammation and microglial activation, highlighting the importance of the microbiota-gut-brain axis (MGBA) in disease pathogenesis. Although the specific literature on this aspect in RTT is scanty, it is known that RTT girls harbour bacterial and fungal microbiota altered as well as subclinical inflammatory status. The AA would discuss the possible role of MGBA in contributing to clinical features of RTT possibly in a brief paragraph.
3) Section 4.3. Mitochondria-Targeted Therapies in RTT. The AA should add the specific literature and and briefly discuss the findings related to the use of metformin, blarcamesine, leriglitozone.
4) Literature should be updated. Only about 26% of the articles are referred to the last 5 years
Minor comments
1) Table 1 legend. Please clarify
2) Section 2.4. The AA should add and discuss the results of mirtazapine treatment in RTT.
Reviewer 2 Report
Comments and Suggestions for Authors
The current manuscript explores the molecular mechanisms of (I) monoamines, (II) immune system and (III) mitochondrial dysregulation in the potential aetiology of Rett Syndrome (RTT).
I have some comments that would strengthen the article:
General comments:
1, I note the authors have stated in the abstract (line 13) and introduction (line 56) that the manuscript is a review (line 1). However, there is no information on how the review was conducted. The authors should consider adding a methodology section to the manuscript to indicate the type of review i.e., scoping/narrative/systematic and add information regarding how the current review was conducted i.e., search terms used, databases searched, how the articles mentioned in the different sections (i.e. 2 etc). and /Tables were identified etc. A figure (like a PRISMA diagram) might also be useful.
2, There are some recent articles that the authors have omitted from the manuscript. These should be included as it would direct readers to the most up-to-date information. Where relevant I have mentioned this in my comments below.
3. Avoid starting a sentence with an abbreviation (lines 36, 48, 82)
Abstract:
4, Line 10-11: The authors should clarify that RTT arises from MECP2 mutations in about 95% of classical and 75% of atypical RTT. This point if important because individuals can have a pathogenic MECP2 mutation without any obvious signs of RTT.
Introduction:
5, Line 28: Reference #1 and #2 are older ones and the authors should consider alluding to the more recent meta-analysis:
Petriti U, Dudman DC, Scosyrev E, Lopez-Leon S. Global prevalence of Rett syndrome: systematic review and meta-analysis. Syst Rev. 2023 Jan 16;12(1):5.
6, The authors should revise the text to do with aspects with the autonomic nervous system (line 35) and sudden death (line 43) especially given that more vulnerable individuals with RTT would be particularly susceptible. In this view, the authors might also want to consider the more recent systematic review:
Singh J, Lanzarini E, Santosh P. Autonomic dysfunction and sudden death in patients with Rett syndrome: a systematic review. J Psychiatry Neurosci. 2020 May 1;45(3):150-181.
7, Line 40 to 41: There are several breathing phenotypes in RTT, and the following more recent article might be better suited here:
Tarquinio DC. et al. The course of awake breathing disturbances across the lifespan in Rett syndrome. Brain Dev. 2018 Aug;40(7):515-529
8, Line 48: This is not quite correct. The authors should consider revising this statement. Because of the lack of association between MECP2 variants and manifestation of RTT symptoms, the diagnosis of RTT remains a clinical one based on core (classical/typical) and supportive criteria (atypical).
9, Line 49 to 52: when describing the regulation of different genes affected by MECP2, the authors should also consider the findings presented in Figure 4 of the following:
Renthal W et al. Characterization of human mosaic Rett syndrome brain tissue by single-nucleus RNA sequencing. Nat Neurosci. 2018 Dec;21(12):1670-1679.
10, Line 56 to 59: The introduction ends rather prematurely, and I would ask the authors to consider adding a more through rationale why the monoamines, cell immunity, and mitochondrial domains were chosen. Each one of these are large areas by themselves and I feel that having a deep dive into each of these areas might be too far reaching. The authors should also have some linking sentences i.e. for the role of monoamines (section 2) to how this relates to immune/inflammatory mechanisms (section 3) in RTT.
11, I also think that while studies have shown some premise for the potential roles of dysregulated monoamine, immune and mitochondrial dysfunction in RTT pathogenesis, the authors should be wary of the causal inference to individuals with RTT.
Section 2
12, Line 74: Please check Table 1. This section is on the roles of monoamines in RTT, so I am unsure why Table 1 is entitled mitochondrial dysfunction in RTT.
13, Line 112: The authors should also consider pertinent examples re: involvement of other neurotransmitter systems and their involvement with breathing dysregulation in RTT. Some information on 5-HT agonists i.e. buspirone could also be mentioned especially given that buspirone has been shown to improve breathing dysregulation and autonomic dysregulation in individuals with RTT. There will be also some overlap with other neurotransmitter systems.
Section 2.4
14, The causal inferences between studies done in RTT animal models and how they relate to studies done in humans should be clarified. For example, the potential for Fluoxetine to increase MeCP2 levels should be tempered (line 195). While this might be true for in animal models the scenario is very different in real world clinical examples. At best Fluoxetine might help to manage the symptoms of anxiety in some individuals with RTT.
15, The following articles are also important when it comes to studies relating to 5-HT and should be considered by the authors:
Guideri F, Acampa M, Blardi P, et al. Cardiac dysautonomia and serotonin plasma levels in Rett syndrome. Neuropediatrics. 2004;35:36–8.
Paterson DS, Thompson EG, Belliveau RA, et al. Serotonin transporter abnormality in the dorsal motor nucleus of the vagus in Rett syndrome: potential implications for clinical autonomic dysfunction. J Neuropathol Exp Neurol. 2005;64:1018–27.
16, Most of section 2.4 is on Fluoxetine, however the authors should also consider the findings done using 5-HT1A agonists, such as tandospirone and buspirone in individuals with RTT.
Section 3
17, Figure 1: Could the authors expand upon Figure 1? How does this diagram relate to the clinical symptoms that manifest in RTT? A box or something similar would be beneficial for the readership.
18, Section 3.4: Could the authors expand on how targeting specific inflammatory mediators could assist in the management of symptoms in individuals with RTT (line 368).
Section 4
19, Line 404: please check the text.
20, Figure 4: I think the statement in lines 406 to 407 should be softened. While there may be perturbations in levels of H202 an SOD in individuals with RTT, how this relates to increased ROS levels and its impact on symptoms of RTT seen clinically is unclear. For example, how reduced AKT/mTOR signaling relates to the symptoms seen in RTT is unknown.
21, Section 4.3: Most of the studies have been done in mouse models and as far as I am aware no clinical trials have shown positive results in ameliorating the mitochondrial deficits in RTT. Hence, this section could be shortened especially to do with coenzyme Q10.
Section 5
22, Line 537: Not much done in terms of patient studies.
23, Line 546: Could the authors expand upon the effective therapies?
Round 2
Reviewer 1 Report
Comments and Suggestions for Authors
The revised version of manuscript is much improved. The manuscript is now suitable for the publication.
Author Response
We thank the reviewer for positive comments.
Reviewer 2 Report
Comments and Suggestions for Authors
The authors have revised the manuscript based on previous suggestions. I have a few points regarding the new text - 1.1 Methods:
1, Line 71: I wanted to double check that if indeed meta-analyses, and systematic reviews were included as part of the search strategy. Usually, these form part of the exclusion criteria when undertaking reviews.
2, Line 74: The authors should consider revising this sentence. As far as I can see a thematic analysis was not done and there is no mention of it in the Methods section.
